# *Triumfetta cordifolia* Gum as a Promising Bio-Ingredient to Stabilize Emulsions with Potentials in Cosmetics

**DOI:** 10.3390/polym15132828

**Published:** 2023-06-26

**Authors:** Michèle N. Fanwa, Catherine Malhiac, Nicolas Hucher, Arnaud M. Y. Cheumani, Maurice K. Ndikontar, Michel Grisel

**Affiliations:** 1Université Le Havre Normandie, Normandie Univ, URCOM UR 3221, F-76600 Le Havre, France; 2Research Unit for Macromolecular Chemistry, Laboratory of Applied Inorganic Chemistry, Faculty of Science, University of Yaounde I, Yaounde P.O. Box 812, Cameroon

**Keywords:** *Triumfetta cordifolia* gum, sustainable substance, emulsifying and stabilizing potential, oil-in-water emulsions, eco-design

## Abstract

The cosmetics industry is searching for efficient and sustainable substances capable of stabilizing emulsions or colloidal dispersions that are thermodynamically unstable because of their high surface energy. Therefore, surfactants are commonly used to stabilize the water/oil interface. However, the presence of a surfactant is not always sufficient to obtain stable emulsions on the one hand, and conventional surfactants are often subject to such controversies as their petroleum origin and environmental concerns on the other hand. As a consequence, among other challenges, it is obvious that research related to new—natural, biodegradable, biocompatible, available, competitive—surfactants are nowadays more intensive. This study aims to valorize a natural gum from *Triumfetta cordifolia* (*T. cordifolia*) as a sustainable emulsifier and stabilizer for oil-in-water (O/W) emulsions, and to evaluate how the nature of the fatty phase could affect this potential. To this end, O/W emulsions were prepared at room temperature using three different oils varying in composition, using a rotor–stator mixer. Resulting mixtures were characterized using optical microscopy, laser granulometry, rheology, pH and stability monitoring over time. The results demonstrated good potential for the gum as an emulsifying agent. *T. cordifolia* gum appears efficient even at very low concentrations (0.2% *w*/*w*) for the preparation and stabilization of the different O/W emulsions. The best results were observed for cocoglyceride oil due to its stronger effect of lowering interfacial tension (IFT) thus acting as a co-emulsifier. Therefore, overall results showed that *T. cordifolia* gum is undoubtedly a highly promising new bio-sourced and environmentally friendly emulsifier/stabilizer for many applications including cosmetics.

## 1. Introduction

Nowadays, cosmetics have become essential, as they have a strong impact on the appearance and personality of individuals. According to the dermatologist Zoe Diana Draelos, the need to feel and be perceived as beautiful is psychologically important to humans, and the use of cosmetics is necessary to improve self-confidence and quality of life [1]. This is why the global cosmetics market is continuously growing (at a compound annual rate of 5% in the 2021–2028 period) [2].

Emulsions are the most common forms of cosmetic products (creams, lotions, hair conditioners). These are colloidal dispersions consisting of two immiscible liquid phases, one of which is finely dispersed in the other in the form of droplets [3]. They are thermodynamically unstable because of their high surface energy. The actors of the cosmetic industry are still looking for efficient and sustainable ingredients that produce stable emulsions while meeting consumer expectations and requirements. Amphiphilic molecules called surfactants, owing both hydrophilic and hydrophobic moieties, are usually used for this purpose. However, the presence of a surfactant is not always sufficient to obtain stable emulsions. The interactions that occur between the components of the mixtures can make them favorable or unfavorable combinations. It has been reported that several oil parameters such as polarity, viscosity and density may affect the properties of emulsions as well as their stability during storage [4,5,6,7,8,9,10]. Thus, for oil-in-water (O/W) emulsions, for example, the emulsifying or stabilizing ability of a surfactant/stabilizer cannot be considered as an absolute and independent parameter but must be assessed according to the type of fatty phase it is supposed to disperse.

Due to the increasing awareness of consumers and current regulations, one of the major drivers of the global cosmetic ingredients market is the growing demand for natural and organic ingredients as substitutes for synthetic ingredients [11]. These ingredients can be of animal or plant origin or of microbial origin through biotechnology. The demand for those of plant origin is further increased by certain lifestyles such as veganism. *Triumfetta cordifolia* (*T. cordifolia*) is a shrub belonging to *Malvacea* family [12,13] and grows in humid areas of tropical Africa [14]. Several species of the same genus have also been reported in America, Asia and Oceania [13,15,16]. *T. cordifolia* grows fast and can reach 2.5 m in one year. It is cultivated either by cutting or by seed with germination rates of 80 to 90%. Fruits with hooked hairs are sometimes dispersed by animal fur, favoring their spread. With this facility of propagation and growth, its demand for industry can be effectively met by agriculture. Additionally, stems can be harvested from the same plant for many years [14]. In cultivated land, *T. cordifolia* and other species of the same genus are common weeds that are difficult to eradicate [14,17,18]. Both leaves and extracts from stem barks are edible [14,18,19]. The stem’s bark, in contact with water, releases a mucilage that has some traditional uses in cooking, where it is used as thickener in some stews [20]. In Cameroon, it is used to cook a sticky soup called “nkui”, a meal eaten mainly on the occasion of births.

Although few in number, previous studies on this gum have reported its monosaccharide composition, which includes neutral sugar units and two uronic acids [21,22]. The composition, reported by Woguia et al., revealed the predominance of rhamnose (41%), then galactose (24%), galacturonic acid (17.5%) and glucuronic acid (10%); the remainder consists of glucose (3%), mannose (2%), arabinose (1.5%) and xylose (1%) [22]. The same sugars were detected by Saidou except for xylose, but it is difficult to exploit the data quantitatively since this last author did not characterize the whole gum sample, but rather the products of several successive extractions. The product of the first extraction, which represents the majority of the gum in the sample, consisted of about 37% of uronic acids including 21.6% galacturonic acid and 15.7% uronic acid. Glucose was the predominant monosaccharide (about 24% in the first extract) [21]. The difference in the quantitative aspect may result from the difference in the place of collection, for example, since they collected the samples in different geographical regions of Cameroon. Indeed, plant materials are often subject to a certain variability due to the influence of several factors such as harvesting season, geographical location, climate, soil composition, age of the plant [23,24].

The above composition reveals that *T. cordifolia* gum has some hydrophilic groups such as -OH and -COO^−^ or -COOH. Saidou also found that it contains some acetylated hydrophobic moieties [21]. Because of the presence of both hydrophilic and hydrophobic groups, *T. cordifolia* gum can be considered amphiphilic and could be expected to have some emulsifying properties. These data, as well as the advantages of the gum in terms of biocompatibility (edible) and availability (easy propagation, fast growth and abundant production), suggest that *T. cordifolia* gum can be a good candidate as a nontoxic bio-emulsifier/stabilizer for the cosmetics industry.

To the best of our knowledge, no work has been reported on the characterization of emulsions based on *T. cordifolia* gum and the impact of the type of the fatty phase on emulsion properties and stability. The emulsifying and stabilizing activities of *T. cordifolia* gum were mentioned by Saidou with a 50/50 *v*/*v* cottonseed oil–water mixture [25]. However, this property was only sketchily highlighted, and the emulsions were not subjected to any characterization: neither the type of emulsion, nor the droplet size distribution, nor the microscopic and macroscopic characteristics of the emulsion, nor the mechanisms involved were provided. Knowledge in this field remains insufficient and scarce.

This work was aimed at showing the potential of *T. cordifolia* gum as a bio-emulsifier and stabilizer for cosmetics on the one hand and studying the effect of the nature of the fatty phase on the gum’s efficiency in order to evaluate with which type of oil the gum potential is optimal on the other hand. For this purpose, the experimental design studied simple O/W emulsions involving three types of fatty phases with different compositions and structures (a blend of hydrocarbon, a blend of fatty acid triglycerides and a blend of glycerides, as shown in Table 1). The three emulsions prepared under identical conditions were characterized using several analytical methods and their properties were compared in order to highlight and understand the effects of the oil type and composition on emulsion formation and stability.

## 2. Materials and Methods

### 2.1. Materials and Reagents

*Triumfetta cordifolia* used in this study was collected in Yaoundé in the Centre region of Cameroon and identified by a botanist at the National Herbarium using the voucher specimen number 41,920/HNC. The gum was extracted from the stem bark according to a well-controlled protocol involving cold extraction using distilled water, filtration, ethanol precipitation and oven drying at relatively low temperature (40 °C) for 24 h. Ultrapure water (R > 18 MΩ-cm, TOC < 1 ppb, bacteria < 1 CFU/mL) was produced by a Barnstead Easypure UV Compact (Thermo Scientific, Illkirch, France). Cosmetic ingredients, namely dehydroacetic acid (and) benzyl alcohol, cocoglycerides, *Helianthus annuus* seed oil (sunflower oil) and *Paraffinum liquidum* (paraffin oil) were provided by Lonza (Basel, Switzerland), BASF (Ludwigshafen, Germany), Rouages (Estillac, France) and Aiglon SAS (Précy-sur-Oise, France), respectively. General characteristics of oils used in this study are listed in Table 1.

Interfacial tension (IFT) is a reliable evaluation of the polarity of oils, being more polar as the IFT decreases [31,32,33]. This is all the more true, as the IFT of oils has been measured versus water, which is a highly polar molecule (due to dipole–dipole interactions). Indeed, the more polar groups there are in an oil, the higher the potential for interaction between the two phases (oil and water), leading to a lower IFT. On this basis and according to the chemical structures of the oils shown in Table 1 (hydrocarbon for paraffin oil, ester for sunflower oil, a blend of ester and polyol for cocoglyceride oil), the relative polarity of these oils with respect to each other can be defined as follows: cocoglyceride oil > Helianthus annuus seed oil > paraffinum liquidum.

### 2.2. Protocol for Emulsion Formulation

Emulsions at an 80/20 *w*/*w* (aqueous phase/fatty phase) ratio studied in this work were formulated according to the protocol described in Figure 1.

*T. cordifolia* gum aqueous solutions were first prepared by sprinkling the required amount of gum powder in ultrapure water while mixing to avoid agglomerates or lumps. The mixture was left under mechanical stirring for 15 h at room temperature (20–25 °C). Then, minimalist O/W emulsions (20/80 *w*/*w*) involving only 4 ingredients (oil, preservative, water and *T. cordifolia* gum) were prepared by cold emulsification using a rotor–stator at a moderate shear rate (6000 rpm) and for a short time (16 min for the whole process).

Although emulsion stability is supposed to increase with gum concentration [25], it is important to investigate the effect of the fatty phase on a nonoptimized system with a relatively low gum concentration in order to better perceive the changes due to the nature of oil. So, emulsions studied in this work were prepared using a gum concentration of 0.2% *w*/*w*. Resulting O/W mixtures were coded F1, F2 and F3 for cocoglyceride, Helianthus annuus seed oil and paraffinum oil, respectively. The composition of these emulsions is shown in Table 2.

It is worth noting that several tests of the gum solutions using different shear systems were performed beforehand to gauge the stability of the gum with respect to shearing. To this end, Ultraturrax^®^ (IKA, Freiburg, Germany), a Silverson high shear mixer (Silverson, Evry, France) and ultrasound cane (Fisher Scientific, Illkirch, France) were used at various speeds and power. The choice was made to use the system that allowed dispersion while preserving the rheological properties of the gum solution. The protocol was repeated 3 times, thus ensuring that it provided fully reproducible emulsions.

### 2.3. Analytical Methods

Several analytical methods were used to monitor emulsions over time. Characteristic periods of monitoring were chosen as follows: day + 1, day + 3, day + 9, day + 18 and day + 31 corresponding to 1, 3, 9, 18 and 31 days of storage after formulation, respectively. The emulsions were stored at 20 °C in a regulated climate room and analyzed over a period of up to 1 month. Rheological measurements were carried out on both gum solutions and emulsions.

#### 2.3.1. Visual Observation

The physical appearance of emulsions was monitored over time using visual observation. For this purpose, photographs of the emulsions were captured at each monitoring date.

#### 2.3.2. Optical Microscopy

The microstructure of emulsions was monitored using a Nikon Eclipse Ni microscope (Tokyo, Japan) with which bright field mode and a 10× objective lens were selected. Nikon NIS-Elements L software, version NIS_L_1.20.00_1419 was used to analyze the micrographs.

#### 2.3.3. Droplet Size Distribution

A laser diffraction particle size analyzer SALD-7500nano (Shimadzu Co., Ltd., Kyoto, Japan) based on static light scattering (SLS) was used to monitor the size distribution of emulsion droplets according to a protocol based on the Fraunhofer theory [34]. Measurements were performed both at the top and the bottom of the emulsions. An absorption range of 0.08–0.20 was attained by dissolving the emulsion sample in ultrapure water prior to measurements. Wing SALD II software was used for data capture. The average diameter of droplets was expressed as the volume-weighted (or de Brouckere) mean diameter D4,3 measured using the software.

#### 2.3.4. Rheology

The flow properties of emulsions were studied using a digital hybrid rheometer DHR3 (TA instruments, New Castle, DE, USA) equipped with aluminum cone–plate geometry (diameter 60 mm; angle 2.006°; truncation gap 53 μm). Flow continuous ramps in the logarithmic mode were performed from 0.001 to 1000 s^−1^ over 300 s at 20 °C.

#### 2.3.5. pH Monitoring

The pH of emulsions was monitored over time using a HACH^®^ SensION™ + PH3 pH-meter (Düsseldorf, Germany). Before measurement, the instrument was calibrated using buffer solutions pH = 4, pH = 7 and pH = 10. Several readings were taken until a stable value was obtained.

#### 2.3.6. Stability Monitoring of Emulsions Using the SMLS Technique

Static multiple light scattering (SMLS) was used to detect particle migration and size variation in liquid dispersions. The system is equipped with two synchronized detectors that record the transmission and backscattering signals. These signals are well interpreted when they are plotted in delta mode, which clearly highlights the destabilization phenomena, as shown in Figure A1 (Appendix A). The correct signal to be used was chosen according to the percentage transmission T (transmission signal if T ˃ 0.2% or backscattering signal if T < 0.2%).

The technology provides high-accuracy measurements of all the variations occurring within a sample. Among other indications, results can be visualized as a unique number called *TSI* (Turbiscan^®^ stability index), calculated as shown in Equation (1).
(1)$$TSI\left(t\right)=\frac{1}{N_{h}}\overset{t_{max}}{\underset{t_{i}=1}{\sum }}\overset{z_{max}}{\underset{z_{i}=z_{min}}{\sum }}\left|BST\left(t_{i},z_{i}\right)-BST\left(t_{i-1},z_{i}\right)\right|$$
where *t_max_* is the measurement point corresponding to the time *t* at which the *TSI* is calculated, z*_min_* and *z_max_* are the lower and upper selected height limits, respectively, $N_{h}=\left(z_{max}-z_{min}\right)/\Delta h$ are the number of height positions in the selected zone of the scan and *BST* is the considered signal (*BS* if *T* < 0.2%, *T* otherwise).

The lower the TSI parameter, the higher the mixture stability; thus, it is characteristic of the destabilization of a given sample. In this work, measurements were carried out using a Turbiscan^®^ Tower (Formulaction, Toulouse, France) equipped with six compartments at a constant temperature of 20 °C.

## 3. Results and Discussion

The ability of *T. cordifolia* gum to form and stabilize O/W emulsions was first tested using a minimalist formula containing 20% *w*/*w* cocoglyceride oil (high polarity and low IFT) or *Helianthus annuus* seed oil (medium polarity and medium IFT) or *Paraffinum liquidum* (low polarity and high IFT) and 80% *w*/*w* aqueous phase containing 0.25% *w*/*w* of gum (for a final gum concentration of 0.2% *w*/*w*). There was no need to adjust the pH since pH of all the emulsions remained stable in the range 4.3–4.8 throughout 31 days, as shown in Table A1 in Appendix B. This range of pH values corresponds to that usually found in cosmetic emulsions to respect the acid–base balance of the skin [35,36,37].

The microscopic aspect of these mixtures (Figure 2) clearly shows droplets evenly dispersed in a continuous medium, confirming the occurrence of an emulsion. *T. cordifolia* gum has a predominant hydrophilic character at the low concentration range investigated. Based on Bancroft’s rule, according to which the continuous phase of an emulsion is the one for which the emulsifying agent has a preferential affinity (it should be hydrophilic for O/W emulsions and oleophilic for water-in-oil emulsions) [38,39], it can be concluded that the type of emulsions formed using *T. cordifolia* gum are O/W emulsions in which oil droplets are dispersed in a continuous water phase.

Emulsion F1 (with cocoglyceride oil) has relatively small droplets, as confirmed by droplet size distributions (Figure 3b) that are bimodal, with an average size of around 27 µm corresponding to a macroemulsion. No change can be noticed when micrographs or particle size distributions between D + 1 and D + 31 are compared, evidence that emulsion F1 is stable. This result is all the more impressive since *T. cordifolia* gum is efficient at very low concentrations (0.2% *w*/*w*) compared to other natural emulsifiers such as acacia gum, which requires very high concentrations (more than 12% or even 20% *w*/*w*) [40,41,42].

However, for emulsions F2 (with *Helianthus annuus* seed oil) and F3 (with *Paraffinum liquidum*), larger oil droplets could be observed compared to emulsion F1 from the initial state, and a creaming phenomenon was evident as oil droplets progressively disappeared from the bottom of these emulsions and migrated to the top. This trend is confirmed by the macroscopic aspects of emulsions (Figure 3a) showing a progressive bottom clarification of emulsions F2 and F3 upon storage. Nevertheless, particle size distributions shown in Figure 3b revealed some differences between emulsions F2 and F3. In the case of F2, there is a progressive depletion of the bottom of the emulsion in large droplets (shift of the distribution towards smaller diameters) and enrichment of the top of the emulsion in large droplets (shift of the distribution towards larger diameters), which confirms once again the creaming mentioned above. In the case of emulsion F3 however, the evolution of the particles size distribution is more complex. While the size distribution of the bottom of emulsion shifted towards smaller diameters owing to droplet depletion, the top of the emulsion underwent both depletion and enrichment in large droplets since in a first step, the top distribution shifted to smaller diameters (red continuous curve at D + 9) and then moved to greater diameters (blue continuous curve at D + 31). Droplet depletion at the top of the emulsion is usually observed in cases of sedimentation where the droplets have a higher density than the continuous phase. In the case of emulsion F3, such a phenomenon is only possible if a phase inversion has occurred on a part of the emulsion, resulting in denser droplets. Analysis of the particle size distribution of the middle of the emulsion could have provided more information. Further analyses are needed to test this hypothesis.

In all cases, whether it is creaming or sedimentation, these destabilization phenomena, obvious in emulsions F2 and F3 but absent in emulsion F1, suggest that the oil properties strongly impacted emulsion stability. Indeed, several oil parameters such as density, polarity, viscosity, surface tension, interfacial tension and droplet radius together govern the breakdown processes occurring in emulsions during storage [4,7].

The major point that can be noted is the difference between initial droplet size of emulsions as shown in micrographs (Figure 2) and confirmed by the particle size distribution in Figure 3 (D4,3 values at D + 1 were 22.9 ± 0.3, 92.8 ± 0.3 and 94.4 ± 0.3 μm for F1, F2 and F3, respectively). The initial droplet size is determined by several factors such as the energy input during emulsification, the viscosity of the dispersed phase, the nature and amount of the emulsifier as well as the properties of the fatty phase such as IFT and polarity [27,43,44]. Then, as other parameters remain identical, the difference in initial droplets size between emulsions F1, F2 and F3 is directly linked to the difference in chemical structures and properties of the fatty phases used, as shown in Table 1. Being a hydrocarbon, paraffin oil does not contain any heteroatoms nor hydrophilic moieties, hence its low polarity and high IFT (54 mN·m^−1^) [27]. Therefore, it cannot contribute to emulsification due to its entire incompatibility towards water. On the other hand, sunflower oil contains some hydrophilic moieties that may spontaneously adsorb at the O/W interface, thus significantly reducing its IFT (27 mN·m^−1^ [29]) and therefore favoring emulsion formation. Finally, cocoglyceride oil, a blend of mono-, di- and triglycerides with the presence of both ester and hydroxyl moieties (with a hydroxyl value of 40–50 mg KOH/g) [26] sharply reduces the IFT to a low value (8.4 mN·m^−1^) [27]. Among the consequences, this last emollient brings a positive contribution to the compatibilization of both immiscible liquid phases. Nevertheless, such a phenomenon remains insufficient to induce self-emulsification in the presence of water with a high shelf-life, since it is not able to provide sufficient interfacial coverage. Experiments performed by mixing cocoglyceride oil and water without any additional emulsifier clearly confirmed that no emulsion could be obtained. As a consequence, in the technical data sheets from the supplier, cocoglyceride oil is claimed only as emollient, not as emulsifier, although it is easily emulsifiable [26]. The results obtained using emulsion F1 are possible only by a relevant contribution of *T. cordifolia* gum to the emulsifying process.

A co-emulsifying effect owing to oil was also reported by Bergfreund et al., which showed that oils with high polarity can act as cosurfactants since their structure contains hydrophilic groups that impart hydrophilic interactions to the aqueous phase through hydrogen bonding and polar π-bonds [33]. The complementary contribution of cocoglyceride oil as a co-emulsifier in emulsion F1 shows that *T. cordifolia* gum has emulsifying properties, but the concentration of 0.2% would be insufficient to emulsify 20% of the fatty phase with a total interfacial coverage, hence the larger initial droplets obtained in emulsions F2 and F3, which do not contain co-emulsifying entities. Therefore, it would be interesting to perform further experiments with higher gum concentrations that attain full surface coverage to obtain emulsions with lower droplet size and enhance stability, even with sunflower and paraffin oils, which do not contain co-emulsifier. Reducing the W/O ratio should also be considered to counteract the creaming observed in these emulsions. Indeed, it has been found that an increase in oil concentration decreases the creaming rate of emulsions due to an increase in emulsion viscosity and packed density of droplets as well as enhancement of interdroplet interactions [45]. Dong et al. also showed that an increase in oil content produces a greater interfacial area between the oil and aqueous phase, resulting in a higher number of emulsified oil droplets with smaller size, thus increasing emulsion stability [46].

The rate of the creaming observed in emulsions F2 and F3 can be commented on based on Stokes’ law (Equation (2))
(2)$$v=\frac{2}{9}\;\frac{\Delta \rho gR^{2}}{\eta }$$
where *v* is the Stokes’ velocity, *R* is the hydrodynamic radius of the droplets, Δ*ρ* is the difference in the densities of the two liquid phases and *η* is the viscosity of the medium, which can be taken as the Newtonian viscosity.

Figure 4 shows the viscosity profiles of emulsions F1, F2 and F3.

The flow profiles of emulsions F2 and F3 were only monitored for 3 days. From the 9th day, these emulsions were no longer homogeneous (Figure 3a), and it would be meaningless to evaluate the flow properties of a nonhomogeneous system. Inversely, emulsion F1 remained homogeneous throughout 31 days, as confirmed by the particle size distribution shown in Figure 3b in which both top and bottom distributions were identical over time, as well as the volume-weighted mean diameter (D4,3), which remained unchanged under storage (Table A2 in Appendix B), thus confirming the stability of emulsion F1.

Figure 4 indicates that all the emulsions have higher viscosity than the aqueous phase. All the curves exhibit, at low shear rates, a plateau region known as the Newtonian domain. As the shear rate increases, a shear-thinning domain appears, resulting from the breaking of the structure and the alignment of molecules in the direction of flow [47]. An analysis of the flow curves using TRIOS software retained the Carreau–Yasuda model as the best fit. The zero shear rate viscosity was taken from the parameters of this model using Equation (3).
(3)$$\eta =\eta_{\infty }+\left(\eta_{o}-\eta_{\infty }\right){\left[1+{\left(\lambda \gamma \right)}^{a}\right]}^{\frac{n-1}{a}}$$
where *η_o_* is the zero-shear viscosity, *η*_∞_ the infinite viscosity, *λ* the consistency (characteristic time), *n* the power law index, and a reflects the transition index describing the transition between Newtonian plateau and power law region) [48].

According to Equation (2), the main parameters controlling the creaming velocity (*v*) of any emulsion are viscosity (η), density difference between oil and the dispersing phase (Δ*ρ*), and radius of the oil droplets (*R*). Emulsion F1 exhibits the lowest values for plateau viscosities, Δ*ρ* and radius *R* compared to emulsions F2 and F3. Low values of Δ*ρ* and *R* tend to increase the stability of the emulsions, while low values of viscosity tend to decrease emulsion stability. Since the difference between the Newtonian viscosities of emulsions was relatively low (viscosities between 7 and 9 Pa·s), the fact that emulsion F1 was the most stable shows that viscosity did not probably have a considerable influence on the stability of these emulsions. Δ*ρ* values are 0.07, 0.09 and 0.17 for cocoglycerides, sunflower and paraffin oils, respectively (a 0.2% *w*/*w* aqueous gum solution with a density of 0.999 g·cm^−3^). The slight difference in Δ*ρ* values, especially between cocoglycerides (F1) and sunflower (F2), showed that the role of density may be fairly limited. The major effect was attributed to the droplets size (D4,3 values at D + 1 were 22.9 ± 0.3; 92.8 ± 0.3 and 94.4 ± 0.3 μm for F1, F2 and F3, respectively). The creaming rate is proportional to the square of the droplets radius, meaning that larger droplets move faster than smaller ones. This is consistent with the rapid creaming of the large droplets in emulsions F2 and F3.

The increase in F1 stability compared to the others, may result from the combined effect of the cocoglyceride oil as cosurfactant and the surfactant properties of the *T. cordifolia* gum as mentioned above, thus enhancing steric stabilization due to high surface coverage of droplets [7,49,50]. Nevertheless, such an effect is limited or not observed in the case of oils with medium and low IFT (medium and low polarity, respectively). In these latter cases, as droplets are not entirely covered by the adsorbed polymer chains, there are bare patches that induce droplet flocculation either by van der Waals attractions between these bare patches (which lead to the droplets’ coalescence if the thin film existing between them breaks down) or by bridging flocculation (the same polymer chain adsorbs onto two or more droplets) [7,51,52,53]. Further investigations will be conducted to ascertain the most plausible mechanism. It is worth mentioning that even for less stable emulsions, more than 60% of the volume of the emulsion remained stable after 31 days (Figure 3a).

The SMLS technique accounts for all the parameters affecting any dispersions including emulsions. Such a tool monitors, in a fairly simple way, the stability of an emulsion and the type of destabilization phenomena occurring in the emulsion.

As an illustration, the variation of backscattering signals of different emulsions measured over a period of 1 month are presented on Figure 5.

The interpretation of these signals, according to the standards, is described in Figure A1 in Appendix A. Although previous results showed no change in the characteristics of the emulsion F1 over time, results from SLMS reveal in this emulsion a tendency for bottom clarification and creaming (Figure 5a). Because these phenomena are due to droplet migration within the emulsion, it would be wise to consider, in a future study, other emulsions based on higher gum concentrations in which the continuous phase, becoming more viscous, would act as a barrier preventing droplet migration with the same ingredients of the emulsion.

In the case of emulsions F2 and F3, which exhibited initially larger oil droplets, weak flocculation and eventually coalescence were coupled to bottom clarification and creaming with, at the same time, a tendency for top clarification in the case of the emulsion F3. It can be observed in Figure 5b that, within a fairly short period of ageing (blue and light green zones of the signals), only flocculation/coalescence (induced either using bare patch van der Waals attractions or polymer bridging between neighboring droplets as described previously) on the one hand and bottom clarification on the other hand first occur, and then the larger coalesced droplets migrate upwards, leading to creaming of emulsion F2, which is consistent with the progression of the D4,3 parameter that increased from 92.8 ± 0.4 to 121.5 ± 0.2 µm at the top while decreasing to 26.9 ± 0.4 µm at the bottom after 31 days of storage (Table A2 in Appendix B). However, in emulsion F3, there is flocculation/coalescence that occurs on the entire height of the tube, clarification at the bottom, but also an alternation of clarification and creaming at the top, as shown in Figure 5c, in which the backscattering signal alternates between positive and negative values as illustrated by arrows corresponding to different periods. Such alternation is congruent with the evolution of the D4,3 at the top of emulsion, which first decreased from 94.4 ± 0.3 µm to 80.4 ± 0.3 µm after 9 days before increasing to 109.9 ± 0.3 µm after 31 days of storage (Table A2 in Appendix B). Both top clarification and creaming are related to the mobility of the droplets according to Stokes’ law, which is a function of the droplet hydrodynamic radius. These observations confirm the hypothesis resulting from the analysis of the droplet size distribution above. It is as though there are two types of droplets in the medium: some, with higher density than the continuous phase and that once coalesced, tend to migrate downwards, and others, with lower density, which tend to migrate upwards. This suggests that the top of the emulsion was undergoing a phase inversion that gave rise to droplets with higher density that may be W/O or multiple (transitional) cohabiting with those having lower density initially present. Further experiments may be required to verify this hypothesis.

Figure 6 reveals that the TSI values over the monitoring period progress in the same direction as the IFT values as TSI (F1) < TSI (F2) < TSI (F3), showing that emulsion F1 was more stable than emulsion F2, which was also slightly more stable than emulsion F3.

The same trend was observed when the different parts of the emulsions (top, middle and bottom) are analyzed separately (Table A3 in Appendix B). It can be concluded that the stability of emulsions based on *T. cordifolia* gum is strongly dependent on the initial droplet size which, in turn, is related to the IFT of the fatty phase.

## 4. Conclusions

The present study aimed to investigate the effect of the nature of the fatty phase on emulsifying and stabilizing potential of *T. cordifolia* gum. For this purpose, simple O/W emulsions textured and stabilized only by *T. cordifolia* gum were formulated using three oils with different compositions, interfacial tensions and polarities. Results revealed that *T. cordifolia* gum efficiently stabilized O/W emulsions, the oil with lower IFT leading to fine and well-dispersed droplets, exhibiting high stability over time. This result is all the more impressive as this remarkable stabilization of the emulsion results from the action of a very low concentration of *T. cordifolia* gum (only 0.2% *w*/*w*) compared to other natural gums such as acacia gums, which require concentrations more than 12% *w*/*w*. Finally, additional experiments with limiting gum concentration showed a partial destabilization resulting from flocculation/coalescence, bottom clarification/creaming as well as top clarification mechanisms, with a significant effect related to initial droplet size, and no significant effect related to oil density.

The overall outcomes of this study clearly highlight that *T. cordifolia* gum can be a promising bio-emulsifier and stabilizer for many industries using oil-based products such as the cosmetics industry.

Further investigations are now intended to elucidate the structure–property relationships of *T. cordifolia* gum and to investigate the emulsifying and stabilizing potential using higher gum concentrations and reduced W/O ratios including interfacial, texture and sensory analyses of more elaborate emulsions close to real cosmetics.

## Figures and Tables

**Figure 1 polymers-15-02828-f001:**
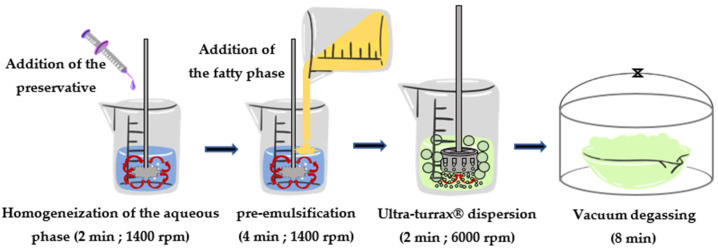
Protocol for the formulation of O/W emulsions based on *T. cordifolia* gum solution.

**Figure 2 polymers-15-02828-f002:**
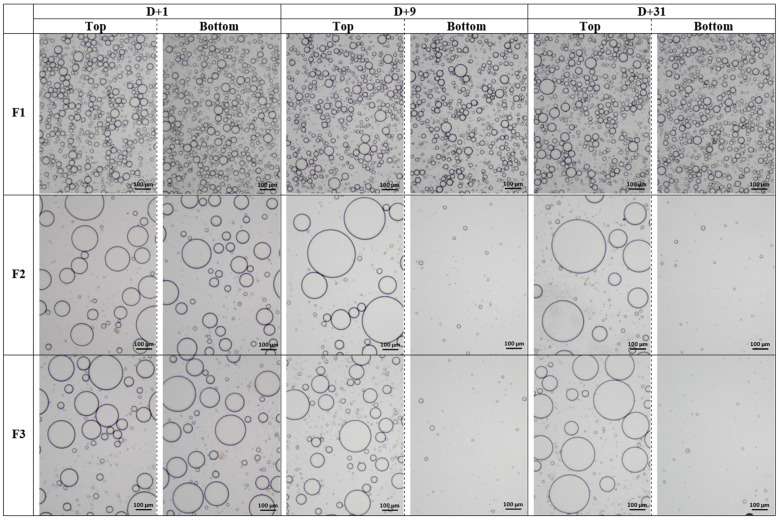
Micrographs of emulsions F1, F2 and F3 stored at 20 °C and monitored for 31 days.

**Figure 3 polymers-15-02828-f003:**
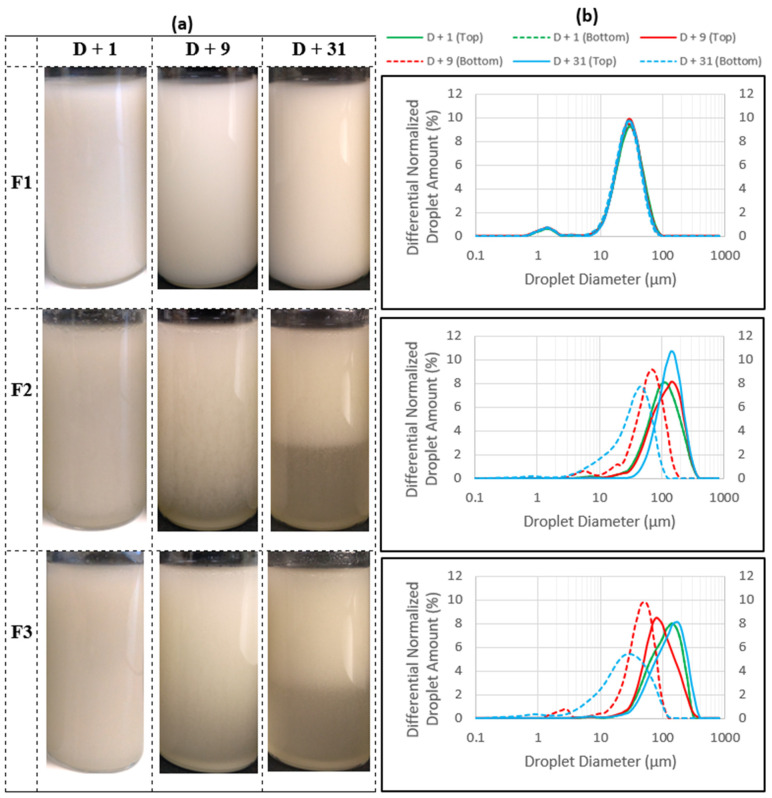
Evolution of the macroscopic aspect (**a**) and the droplet size distribution (**b**) of emulsions F1, F2 and F3 over 31 days.

**Figure 4 polymers-15-02828-f004:**
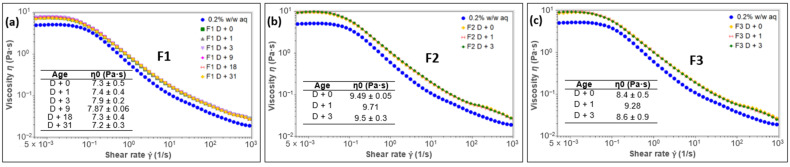
Flow curves of emulsions F1 (**a**), F2 (**b**) and F3 (**c**) monitored until the system started to become nonhomogeneous.

**Figure 5 polymers-15-02828-f005:**
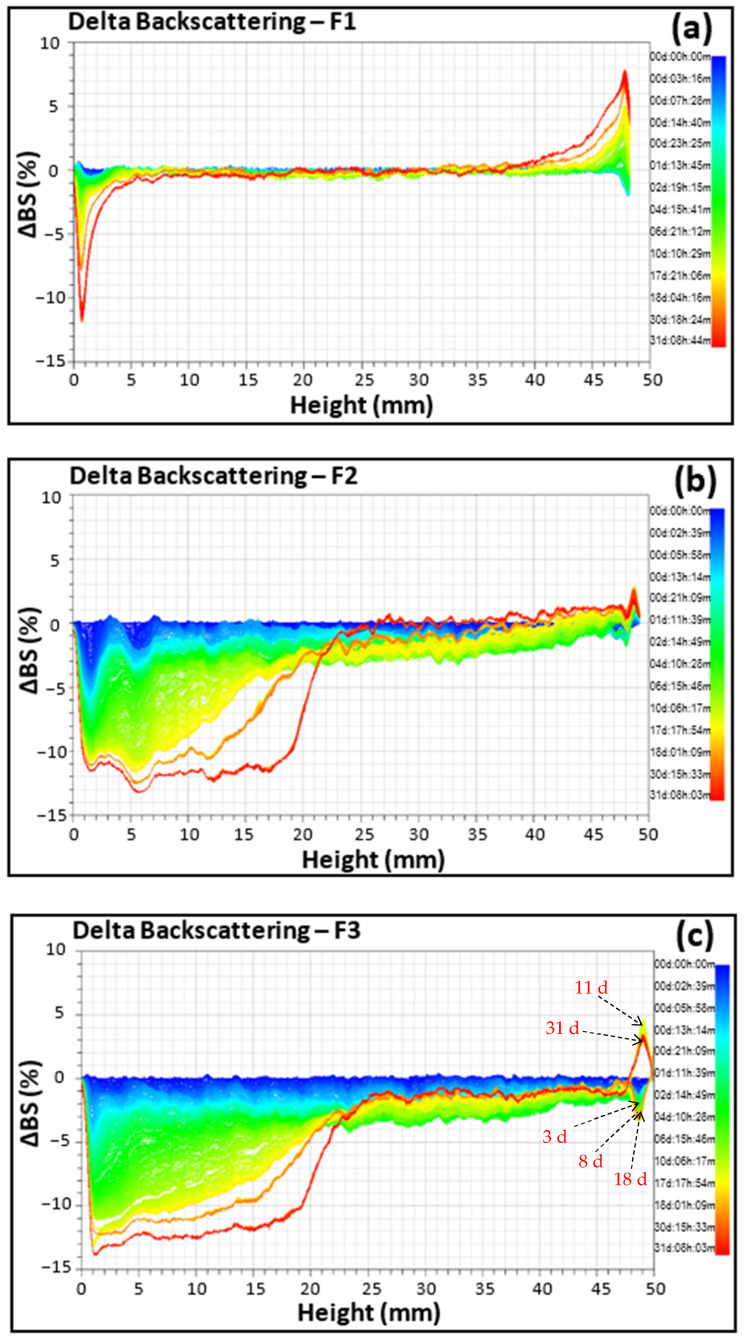
Delta backscattering signals of emulsions F1 (**a**), F2 (**b**) and F3 (**c**).

**Figure 6 polymers-15-02828-f006:**
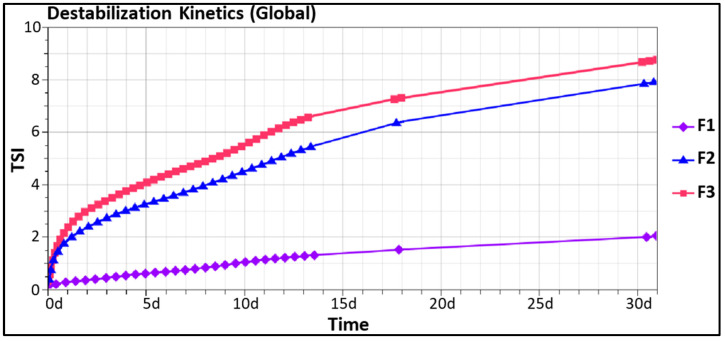
Turbiscan^®^ destabilization kinetics of emulsions F1, F2 and F3 over 31 days.

**Table 1 polymers-15-02828-t001:** Characteristics of oils used as fatty phases.

Trade Name	INCI Name	Composition	Density at 22 °C (g·cm^−3^) *	Viscosity at 20 °C (mPa·s)	Surface Tension at 25 °C (mN·m^−1^)	Interfacial Tension versus Water at 25 °C (mN·m^−1^)
Myritol 331	Cocoglycerides	Blend of mono-, di and triglycerides ** R-OOC-CH_2_-CH(OH)-CH_2_-OH R-OOC-CH_2_-CH(COOR)-CH_2_-OH R-OOC-CH_2_-CH(COOR)-CH_2_-COOR	0.93	43–48 [26]	29.54 ± 0.01 [27]	8.37 ± 0.36 [27]
Sunflower oil ***	*Helianthus annuus* seed oil	Fatty acid triglycerides R-OOC-CH_2_-CH(COOR)-CH_2_-COOR with residual free fatty acids	0.91	63.9 [28]	35.1 ± 0.34 [29]	27.1 [29]
Paraffin oil Codex AAB2	*Paraffinum liquidum*	Alkanes -R	0.83	32 [30]	30.02 ± 0.02 [27]	54.00 ± 0.56 [27]

* Data obtained from this study. ** Hydroxyl value: 40.0–50.0 mg KOH/g [26]. *** High oleic sunflower oil was used as it contains less polyunsaturated fatty acids hence it is more stable from oxidation.

**Table 2 polymers-15-02828-t002:** Composition of emulsions.

Phase	Ingredient (INCI Name)	Function	Weight Percentage (%)
Aqueous phase	Ultrapure water	Solvent	78.8
	Polymer (*T. cordifolia*)	Emulsifying, stabilizing and texturizing agent	0.2
	Dehydroacetic acid (and) benzyl alcohol	Preservative	1.0
Fatty phase (code of emulsion)	Cocoglycerides (F1)	Emollient	20.0
	*Helianthus annuus* seed oil (F2)		
	*Paraffinum liquidum* (F3)		

## Data Availability

Data are contained within this article.

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
