# Peer review of "Triumfetta cordifolia Gum as a Promising Bio-Ingredient to Stabilize Emulsions with Potentials in Cosmetics"

_polymers, 2023, doi:10.3390/polym15132828_

Round 1

Reviewer 1 Report (Previous Reviewer 2)

I think that this new version has improved and it can be published after some minor revision. There is only one error that has to be corrected:

- In line 250 reference numbers are not correct, they should be [38, 39], because references [36, 37] are at lines 242-243

Author Response

Please find the author's response in the word file.

Reviewer 2 Report (New Reviewer)

In the paper entitled Triumfetta cordifolia gum as a promising bio-Ingredient to stabilize emulsions with potentials in cosmetics, the authors reported that a natural gum from Triumfetta cordifolia as a sustainable emulsifier and stabilizer for oil-in-water (O/W) emulsions, and evaluated how the nature of the fatty phase could affect this potential. I have some comments to the authors.

1.      Spaces are missing between number and °C, and all of the manuscript is missing spaces between the numbers and the units of temperature, such as Line124.

2.      Fig. 3: The word “particle” in the figure and title should be corrected to “droplet”.

3.      It is recommended to use d3,2 or d4,3 instead of the mean diameter values to analyse whether the diameter of droplets has changed significantly during storage in order to analyse whether the sample is stable.

4.      F2 and F3 exhibit creaming phenomenons, with variations in droplet size between the top and bottom layers. It might be worth considering increasing both the internal phase fraction and higher gum concentrations to reduce the aqueous phase.

Author Response

Please find the author response in the word file.

This manuscript is a resubmission of an earlier submission. The following is a list of the peer review reports and author responses from that submission.

Round 1

Reviewer 1 Report

The present manuscript describes the stability of three emulsions prepared with 20 wt. % cocoglycerides/ sunflower oil or paraffin oil dispersed in 0.2 % (w/w) aqueous solution of Triumfetta cordifolia gum. The main result of the study is that the emulsion with cocoglycerides is shown to be much more stable compared to the other two (with sunflower or paraffin oil). Thus, the authors conclude that the T. cordifolia gum is a promising new ingredient for preparation of cosmetic emulsions, and also – that this gum is more prominent for stabilization of emulsions containing highly polar oils as dispersed phase compared to oils with lower polarity.

However, these conclusions are not really unambiguously defended by the results presented in the paper. Besides the fact that only 3 systems have been explored (with undisclosed reproducibility of the presented results), there are several crucial flows in the analysis made by the authors:

-          The effect of the interfacial tension in the different systems has not been considered. However, it is very well known that the outcome of the emulsification process depends primarily on this factor, as well as the energy input to the system. Hence, the differences in the initial drop sizes observed in the 3 samples are most probably explained by different IFT of the systems.

-          The stability of emulsions with very different initial drop sizes is used to make conclusions about the effect of the oil polarity. In my opinion, the destabilization effect seen in samples with sunflower oil and paraffin oil is mainly related to the initial drop size and the presence of additional emulsifiers in the cocoglycerides, see next point.

-          The cocoglycerides per se are known to be a mixture of mono-, di- and triglycerides derived from coconut oil. Although, the authors have not specified what is the trade name of the cocoglycerides purchased from BASF, unless they have taken special procedure to remove the mono- and di-glyceride molecules from the cocoglycerides, these are present in the prepared emulsions and can therefore act as emulsifiers in the sample as well. Hence, it is very probable that the smaller sizes observed in this sample are not related to the emulsifying properties of the gum, but to these of mono- and di-glyceride molecules.

A careful consideration of all these points, along with investigation of the effect of emulsification procedure, gum concentration and clarification of all mechanisms that are presently claimed by the authors to need further investigation (see for example lines 246-255; 327-330; 347-352..) is needed in order for the paper to become acceptable for further consideration and publication.

There are several sentences which need to be revised - see for example - lines 44, 273, 287, 318.

Reviewer 2 Report

This work presents an initial study on the potential of a gum obtained from a plant (Triumfetta cordifolia) in order to be used as an environmentally friendly surfactant in oil-in-water emulsions. Oil-in-water emulsions stabilized with the gum are prepared with three oils with different polarities and they are characterized by optical microscopy, laser granulometry, rheology, pH and stability over time. The work is focussed on the study of the effect of oil polarity on the emulsion formation and stability. The authors state that the Triumfetta cordifolia gum shows efficiency at very low concentrations (0,2% w/w) compared with other natural gums as acacia gum.

I think that the work is interesting on one hand because of the efficiency of the gum as emulsifier and on the other hand because of the properties of the plant, described by the authors, its cultivation can be sustainable in some areas of the world.

I think that the manuscript can be published after major revision.

-       i)    A table with the wt% of components of each of the mixtures should be included (wt% of fatty phase, wt% water and wt% of gum,) and the code.

-         ii)The main object of the paper is to study the effect of the polarity of the oil phase. The author describe the polarity of the oils employed as high, medium and low. However, I think that some quantitative information should be given. Therefore, authors should include surface tension values of the oils

-          iii) A table should be included with the properties of the oils: density, viscosity and surface tension.